# Mixed-Etiology Restrictive Cardiomyopathy (Desminopathy and Hemochromatosis) with Complex Liver Lesions

**DOI:** 10.3390/genes13040577

**Published:** 2022-03-24

**Authors:** Yulia Lutokhina, Olga Blagova, Alexander Panferov, Vsevolod Sedov, Evgeniya Kogan, Tatiana Nekrasova, Alexander Nedostup, Elena Zaklyazminskaya

**Affiliations:** 1V.N. Vinogradov Faculty Therapeutic Clinic, I.M. Sechenov First Moscow State Medical University (Sechenov University), 119991 Moscow, Russia; blagovao@mail.ru (O.B.); a_panferov@mail.ru (A.P.); avnedostup@mail.ru (A.N.); 2Department of Radiology, I.M. Sechenov First Moscow State Medical University (Sechenov University), 119146 Moscow, Russia; vps52@mail.ru; 3Department of Pathology, I.M. Sechenov First Moscow State Medical University (Sechenov University), 119991 Moscow, Russia; koganevg@gmail.com (E.K.); petrovna257@rambler.ru (T.N.); 4Laboratory of Medical Genetics, B.V. Petrovsky Russian Research Center of Surgery, 119991 Moscow, Russia; helenezak@gmail.com

**Keywords:** restrictive cardiomyopathy, desmin, *HFE1* heterozygous hemochromatosis, paroxetine, retinol

## Abstract

A 28 year-old male with restrictive cardiomyopathy (RCM) and endocardium thickening, conduction disorders, heart failure, and depressive disorder treated with paroxetine was admitted to the clinic. Blood tests revealed an increase in serum iron level, transferrin saturation percentage, and slightly elevated liver function tests. Sarcoidosis, storage diseases and Loeffler endocarditis were ruled out. Mutations in desmin (*DES*) and hemochromatosis gene (*HFE1*) were identified. Liver biopsy was obtained to verify the hemochromatosis, assess its possible contribution to the RCM progression and determine indications for treatment. Biopsy revealed signs of drug-induced injury, subcompensated heart failure, and hemosiderin accumulation. Thus, even if one obvious cause (desmin mutation) of RCM has been identified, other less likely causes should be taken into consideration.

## 1. Introduction

The term “restrictive cardiomyopathy” (RCM) includes a wide range of diseases that vary in etiology and pathogenesis. The main common feature of these various conditions is marked diastolic dysfunction. The current RCM definition is more descriptive rather than categorical. According to the Cardiomyopathy Classification of 2008, restrictive cardiomyopathy is defined as lesion with restrictive ventricular physiology in the presence of normal or reduced diastolic volumes (of one or both ventricles), normal or reduced systolic volumes, and normal ventricular wall thickness [1].

The prevalence of RCM is currently unknown. It is considered to be the rarest type of cardiomyopathy [2]. In the European Classification of cardiomyopathies, it is proposed to distinguish between familial (caused by mutations in the genes of sarcomeric proteins and other genes; amyloidosis (TTR, apolipoprotein), haemochromatosis, Fabry disease, glycogen storage diseases, elastic pseudoxantoma) and non-familial (AL-amyloidosis, sarcoidosis, systemic sclerosis, carcinoid heart disease, metastatic lesion, radiation and anthracyclines induced, endocardial fibrosis) forms of RCM [1]. Another RCM classification is based on the myocardial and endomyocardial causes of RCM [3]. Treatment of RCM includes not only congestive heart failure (CHF) management [4] but as well treatment of the underlying cause (if identified).

We report a case of mixed-etiology RCM combined with liver lesions. Written informed consent was obtained from the patient for publication of this case report and any accompanying images.

## 2. Materials and Methods

A 28 year-old male was referred to the clinic in February 2019 with complaints of episodes of greyout in the eyes, accompanied by a decrease of the heart rate to 40 BPM, and reduced tolerance for physical exertion. The patient underwent clinical examination including collection of venous blood samples for DNA-diagnostics, blood tests (full blood count, biochemical panel and international normalized ratio, INR), electrocardiography (ECG), 24-h ECG monitoring, echocardiography (ECHO), chest computed tomography (CT), liver magnetic resonance imaging (MRI), and liver biopsy (hematoxylin and eosin staining, Van Gieson staining, Perles staining).

Whole exome sequencing (WES) for the proband’s DNA was performed using a SureSelect All Exons V7 library preparation kit (Agilent Technologies, Santa Clara, California, USA) followed by next-generation sequencing on an Illumina platform (NextSeq 550, Illumina, San Diego, CA, USA). Reads were aligned to the human genome build GRCh37/UCSC hg19 and the variant calling with an automatic module EMSEMPLE_VEP with the following analyze of the sequence variants using a custom-developed bioinformatics pipeline. Confirmation of genetic findings in proband was performed by capillary Sanger resequencing on an ABI 3730XL DNA Analyzer according to the manufacturer’s instructions (Thermo Fisher Scientific, Waltham, MA, USA). The direct capillary Sanger resequencing with alternative oligoprimers was performed for the coding exons and flanking 100 bp of intronic areas of the *HFE* gene to avoid an occasional loss of the second rare variant on NGS testing.

Pathogenicity assessment of all variants confirmed by Sanger re-sequencing was performed according to ACMG (2015) criteria (on behalf of the ACMG Laboratory Quality Assurance Committee) [5].

The study was performed in concordance with the Declaration of Helsinki in its current form and approved by the Sechenov University Local Ethics Committee (protocol № 11–15 from 16 February 2019), Moscow, Russia. Voluntary informed consent was obtained from the patient.

## 3. Results

Significant events in the patient’s family history were found in the mother’s male lineage: a fatal stroke at 50 years of age in an uncle; the death of the grandfather at 53 years of age. No other family members were affected by RCM. The patient did not smoke, drink alcohol, or follow any special diet. He has worked out regularly at the gym (moderate exercise). From 23 years of age (**2014**), complete right bundle brunch block (RBBB) on the ECG and frequent premature supraventricular beats (PSBs) appeared. Since **2017**, there have been episodes of transient II degree, type 2 AV block, accompanied by presyncope. From this period, the patient started taking paroxetine 20 mg/day due to depressive disorder. In **November 2018** there was an episode of CHF decompensation with ascites, hydrothorax, hydropericardium, which required hospitalization in the emergency department. A bit later, a transient III degree AV block was registered (Figure 1A). The **ECHO** revealed moderate hypertrophy of the left ventricle (LV, septum, and posterior wall 1.4 cm), left atrium (LA) dilatation up to 112 mL and diastolic dysfunction with a restrictive filling pattern. Ejection fraction (EF) of LV was preserved (66%). To rule out the glycogen storage disease, the activity of lysosomal enzymes was assessed: all indexes were normal. As well, a **cardiac MRI** was performed: LV EF 53%, interventricular septum 1.2 cm, LA 200 mL, diffuse accumulation of contrast with thickened endocardium (3–4 mm) of both left and right ventricles, site of intramyocardial late gadolinium enhancement (LGE) in LV were observed.

On examination in our clinic, the patient had a regular pulse (110 BPM), BP was 140/80 mm Hg. A single hyperpigmentated site (3 cm) was on the forearm skin. Physical examination revealed enlargement of the liver (+2 cm from the costal margin). No signs of pulmonary congestion or peripheral edema were found.

**Blood tests** including full blood count, biochemical panel (including creatine kinase) and INR were normal except for an increase in serum iron level (40.9 μmol/L, normal range 9–31), transferrin saturation percentage (85.2%) and slightly elevated liver function tests (AST 39 U/L, ALT 63 U/L). The **ECG** showed bifascicular block (Figure 1C). The **ECHO** confirmed the presence of restrictive cardiomyopathy with hyperechogenic and layered LV myocardium, which could be the sign of storage disease (Figure 2, Appendix A). **Chest CT** was normal, without any signs of mediastinum lymphadenopathy. **Liver MRI** confirmed hepatomegaly.

Early onset of symptoms, their gradual and steady progression, combined with features of clinical course made the diagnosis of primary RCM the most probable. The whole exome sequencing was performed. A mutation in the sarcomeric gene desmin (*DES*) in the heterozygous state was found. Missense variant chr2:g.220290456C>T (ENST00000373960.3:c.1360C>T, ENSP00000363071.3:p.Arg454Trp) in the *DES* gene affects the tail region of desmin protein in a direct neighborhood of CRYAB-interacting domain (aminoacids 438–453) [6]. This variant was shown to disturb severely filament-formation competence and filament-filament interactions, and to increase interaction with CRYAB protein (PS3 strong criterion) [7], not found in gnomAD genomes (PM2_Supporting); its pathogenic computational verdict was based on 12 pathogenic predictions vs. no benign predictions (PP3_Supporting criterion), and ClinVar classifies this variant as Pathogenic with two stars (multiple consistent, 10 submissions) (PP5_Supporting). Summarizing this evidence, we classify this variant as Likely Pathogenic (Class IV). DNA samples from parents were unavailable and the origin of the mutation (de novo vs. inherited) was not tested.

At the same time, the prominent RCM phenotype required ruling out its secondary forms. Amyloidosis was highly unlikely, due to the age of the patient and early onset of symptoms. Cardiac sarcoidosis was more likely among the secondary forms (a typical pattern of LGE in MRI, RBBB combined with severe AV block), but chest CT demonstrated no signs of sarcoidosis. There is an accumulation of contrast with thickened endocardium by MRI but the absence of eosinophilia in the blood and intraventricular thrombosis made Loeffler endocarditis unlikely. Blood tests showed a marked increase in serum iron level and transferrin saturation and a moderate increase of liver function tests, together with liver enlargement and site of skin hyperpigmentation it made hemochromatosis plausible. In the *HFE1* gene, a recessive mutation, typically associated with hemochromatosis was found, but in heterozygous state: missense variant chr6:g.26093141G>A (ENST00000357618.5:c.845G>A, ENSP00000417404.1:p. Cys282Tyr). This variant is a well-known common mutation (prevalence is 0.03825 in gnomAD genomes) [8]. It affects the Alfa-3 region of Hereditary hemochromatosis protein (aminoacids 206–297) and posttranslational modification of the protein disturbing disulfide bond between 225Cys and Cys282 aminoacids [9] (Criterium PS3_Strong and PM1_Strong); pathogenic computational verdict based on 7sevenpathogenic predictions vs. 3 benign predictions (PP3_Supporting); it was classified as pathogenic in more than 10 articles and many ClinVar submitters (PP5_Strong). Taking all the evidence, we classify this variant as Pathogenic (Class V). As it is in a heterozygous state, it does not fully confirm but also does not deny the presence of hemochromatosis.

The patient had absolute indications for implantation of a pacemaker. However, because of unpredictable progression of CHF and as a consequence of highly possible ventricular rhythm abnormalities, a cardioverter defibrillator (ICD) was implanted. CHF treatment was initiated (bisoprolol up to 5 mg/day with subsequent withdrawal, torasemid 5 mg/day and spironolactone 25 mg/day). A diet with a strict restriction of iron was recommended.

The patient was admitted again in **December 2019** to correct the treatment. He was doing well except for episodes of irregular heartbeat. The physical examination revealed hepatomegaly (+3 from the costal margin), the consistency of the liver was firm, palpation was painless. The **ECHO** did not show any significant dynamics. **Holter monitoring** (Figure 1B) revealed 7700 PSBs. Lappaconitine hydrobromide (class IC) 75 mg/day was administered. The number of PSBs reduced to 1500/day. **Blood tests** showed an increase in hemoglobin to 17.5 g/dL, erythrocytes 5.8 × 10^6^/L, hematocrit 51%, a moderate increase in total and direct bilirubin, normalization of serum iron level and transferrin saturation percentage. The last, however, did not allow us to rule out hemochromatosis. In addition, a significantly enlarged and firm liver in the absence of other signs of right-sided heart failure could be considered to be a sign of hemochromatosis.

**A liver biopsy** was obtained to verify the diagnosis of hemochromatosis, assess its possible contribution to the RCM progression and determine indications for treatment. The biopsy revealed three groups of morphological changes (Figure 3): (1) The nuclei of several hepatocytes were large with optically empty vacuoles. Fragments of the cytoplasm of some hepatocytes resembled ground glass (hyperplasia of the granular endoplasmic reticulum). Moderate focal hydropic and balloon dystrophy of hepatocytes of zones 2 and 3 of the acinus and moderate diffuse fine fatty dystrophy of hepatocytes (up to 40%) without zonal predominance (++) were observed. In the cytoplasm of hepatocytes of acinus zones 2 and 3, there were moderate amounts of small and large greenish-brown Pearl’s-negative granules: bilirubin. Diffuse steatosis (++), bilirubinostasis and cholangiopathy, and signs of hepatocytes hyperfunction are consistent with drug exposure to the organ; (2) dilation of centrilobular sinusoids, central vein sclerosis and capillarization of adjacent sinusoids may be the consequence of prolonged CHF; (3) mild hemosiderin accumulation is a sign of hemosiderosis (+).

Two years later, the patient remains relatively stable. He is now on the waiting list for heart transplantation in the case of deterioration or heart failure.

## 4. Discussion

RCM etiology could be primary (including familial forms), acquired, or related to systemic diseases [1]. Because RCM is the rarest cardiomyopathy, the genetic spectrum of mutations related to RCM is still unclear [10]. However, there are interesting data about the genetic basis of idiopathic RCM in 32 unrelated patients: the disease-causing mutation was identified in 19 (79%) of those cases, and 3 of 19 patients had a mutation in *DES* [11].

Desmin encodes intermediate filament protein. Pathological desmin conglomerates in the z-lines area lead to apoptosis, fibrosis and finally to CHF development. Conduction disorders, including RBBB and AV blocks, are typical for desminopathy [12]. Previously, the case of RCM was described in a 19-year-old man with a homozygous missense mutation in *DES*: like our patient, the first manifestation of disease was complete RBBB, and three years later the 3d degree AV block developed [13].

As for the reason for the liver lesions, we considered hemochromatosis contributing both to RCM and liver injury. Our patient is heterozygous by only one *HFE1* gene mutation. It should be the second mutation in the second allele to have a “gold standard” confirmation of the diagnosis of a recessive hemochromatosis, but we failed to find it in two rounds of sequencing by two independent methods. However, all sequencing approaches have their technical limitations, and the negative result of genetic testing has no exclusive power. The second genetic variant might be located deeply in intronic or regulatory areas or might be technically “invisible” for NGS or Sanger sequencing methods (i.e., small deletion/insertion of the whole coding exon(s)). Moreover, there is data in the literature about iron overload even in *HFE1* heterozygous carriers [14]. The liver biopsy revealed the presence of hemosiderin accumulation in liver tissue. Features of subcompensated CHF were described as well, which do not contradict our diagnostic concept. Moreover, the liver biopsy revealed signs of drug-induced liver lesions. The patient is known to have been taking paroxetine 20 mg/day for two years. At least eight cases of liver damage have been reported while taking this drug [15]. Paroxetine increases serotonin concentration both in the central nervous system and in blood [16]. It may simulate the situation in carcinoid heart disease and presumably lead to thickening of the endocardium, which we observe in our patient according to MRI. This, along with hemochromatosis, may add a secondary component to the RCM. Moreover, from the patient’s history, we found that he had previously received high doses of retinol to treat acne. In addition, hypervitaminosis A could cause portal hypertension, fibrosis, and even cirrhosis [17,18]. Hemosiderin accumulation could be secondary as well as a consequence of drug-induced liver injury or chronic venous congestion in the liver due to CHF [19].

Therefore, regardless of the genetically verified primary RCM, this patient may have secondary reasons that may worsen the restrictive phenotype and lead to liver damage, such as hemochromatosis or prolonged paroxetine administration. A combination of these factors could lead to an earlier onset of symptoms and a more prominent clinical course.

## 5. Conclusions

This clinical case shows that RCM etiology could be combined and even if one obvious reason has been identified successfully, it is worthwhile to take into consideration other possible but less likely causes. The same diagnostic approach is appropriate for interpretation of hepatic injury in a patient with CHF, which could have mixed nature as well.

## Figures and Tables

**Figure 1 genes-13-00577-f001:**
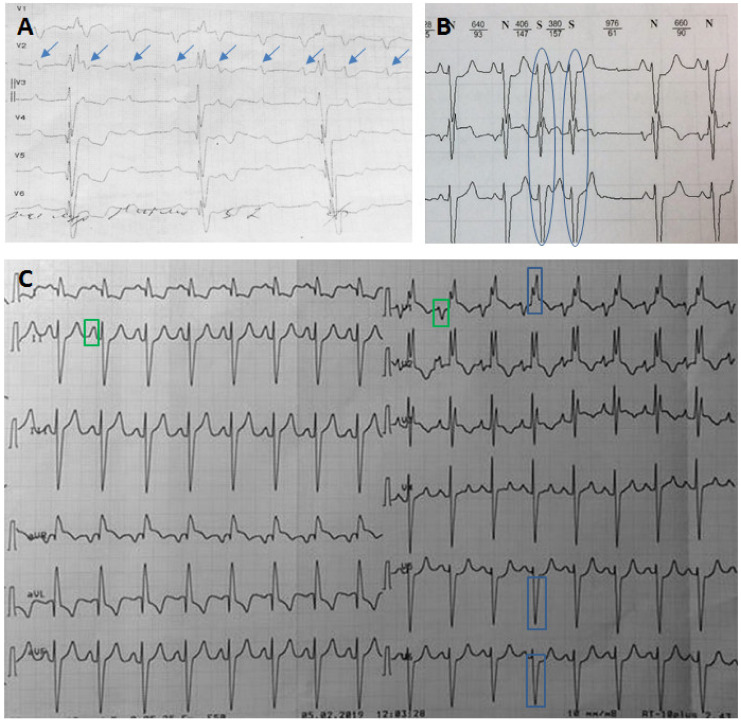
Electrocardiograms (ECGs) of the patient. (**A**) ECG fragment with complete AV block (p-waves are indicated by blue arrows). (**B**) A fragment of Holter ECG monitoring with frequent premature supraventricular beats (premature complexes are indicated by blue ovals). (**C**) ECG of the patient on admission. Sinus tachycardia (100 beats per minute), right bundle branch block (indicated by blue rectangles) combined with left anterior fascicular block (left axis deviation). Signs of enlargement of both atria (indicated by green rectangles).

**Figure 2 genes-13-00577-f002:**
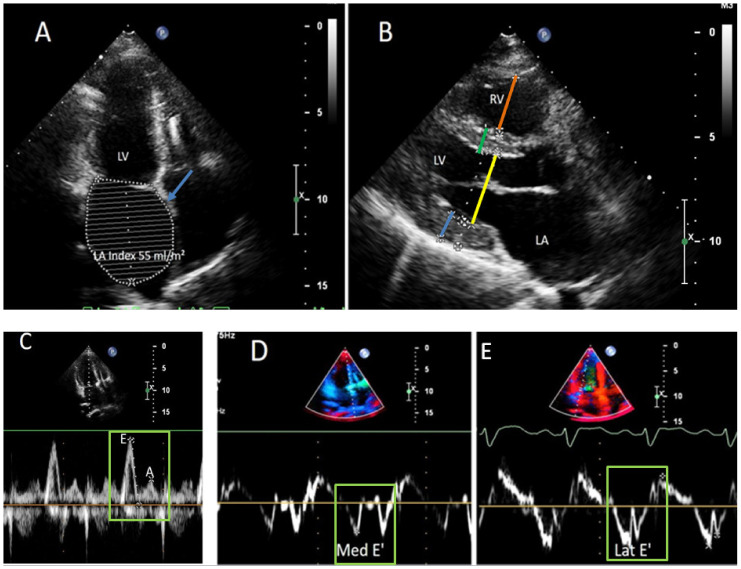
Echogardiograms of the patient on admission. (**A**) Apical two chambers view (left atrium (LA, indicated by blue arrow) volume index 55 mL/m^2^). (**B**) Parasternal long-axis view (ventricular septum 1.2 cm (green line); left ventricular posterior wall 1.2 cm (blue line); right ventricular mid diameter 2.4 cm (orange line); left ventricular diameter 3.6 cm (yellow line). (**C**) Transmitral flow pattern (E: 113 cm/s; A: 39 cm/s; E/A 2.8; dt: 90 msec, flows are indicated by green rectangle). (**D**,**E**) Tissue Doppler (E_med_: 10 cm/s; E/E_med_: 10.6; E_lat_: 10–11 cm/s; E/E_lat_: 10.8; flows are indicated by green rectangles).

**Figure 3 genes-13-00577-f003:**
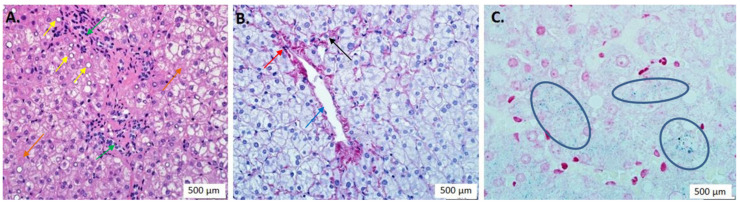
Results of the liver biopsy. (**A**) Hematoxylin and eosin staining. Moderate diffuse microvesicular steatosis (++, indicated by yellow arrows), parenchymatous bilirubinostasis (indicated by orange arrows) and cholangiopathy with proliferation of small ducts (indicated by green arrows). (**B**) Van Gieson staining. Dilation and full-blooded centrilobular sinusoids (indicated by red arrow), moderate central vein wall sclerosis (indicated by blue arrow) and capillarization of adjacent sinusoids (indicated by black arrow). (**C**) Perles staining. Initial signs of hemosiderin accumulation in liver tissue (in blue ovals) — hemosiderosis (+). The number of pluses reflects the intensity of the sign.

## Data Availability

The data presented in this study are available on request from the corresponding author.

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
