# Peer review of "Mixed-Etiology Restrictive Cardiomyopathy (Desminopathy and Hemochromatosis) with Complex Liver Lesions"

_genes, 2022, doi:10.3390/genes13040577_

Round 1

Reviewer 1 Report

The manuscript describes the case of a 28-yo male with RCM, endocardial thickening and conduction disease, depressive disorder treated with paroxetine, and increased levels of serum iron and transferrin saturation. Authors performed exome sequencing and identified "Mutations in desmin (DES) and hemochromatosis gene (HFE1)". The authors conclude that the obvious cause of RCM is DES mutation and that "other less likely causes should be taken into consideration".

Comments for the authors

Major missing information:

The DES variant.

The HFE1 variant.

The lack of description of the variant in DES makes it difficult to evaluate the case.

Family information is missing. Were other family members affected by RCM?

Was skeletal muscle involved? (add levels of sCK)

Liver biopsy revealed signs of drug-induced liver lesions: please describe specific features.

“Skin hyperpigmentation made hemochromatosis plausible”: please show

The sentence: "It does not fully confirm but as well does not deny the presence of hemochromatosis". Having conducted an exome sequencing analysis, the authors will have excluded pathogenic variants in HJV, SLC40A1, TFR2, FHT1, HAMP, BMP2. Is that correct?

Reviewer 2 Report

The manuscript by Dr. Lutokhina and colleagues describes the case of a young patient affected by RCM, in which a DES mutation has been found. However, additional aetiologies can be recognised beyond the classical genetic one, in inherited hemochromatosis and/or drug-induced hepatitis.

Despite the concept of multiple causes is novel and interesting, this work lacks some essential information, crucial for the judgement of the case report.

  • The description of the mutations is mandatory, with the pathogenic characterization (according to American College of Medical Genetics and Genomics Guidelines) and the variant frequency in the general population. The method for the genetic screening needs to be specified, with details on the variant calling technique and which genes were read. In addition HFE1 gene variant has been found in one allele only, while hemochromatosis is a recessive disease. Please address this in the discussion.
  • The described case would have benefit from additional exams, such endomyocardial biopsy, to confirm hemochromatosis, to exclude sarcoidosis, glycogen storage disease and to visualize eventual inflammatory infiltrates.
  • What are the typical signs of hepatic drug- induced injury vs. genetic hemosiderin accumulation?

Further, different points could be improved:

  • Please refer to more up to date guidelines, such as: doi:10.1093/eurheartj/ehab368
  • What does (2+), (+1) mean in the legend of figure 2?
  • Scale bar in figure 2 is not well evident
  • Please detail if figure 2D has two panels or add figure 2E
  • Please make the figures more legible even for not-experts (by adding arrows or indications of measures)
  • Please detail alcohol intake and alimentary – physical exercise habits. Maybe they are more relevant then the use of retinol for acne…
  • How did you check absence of eosinophilia if you don’t have a cardiac biopsy?
  • Please rephrase to make easily readable and understandable the sentences of lines 37-43. In particular, what do you mean for “Another variant of RCM classification depends on the mechanism of myocardial stiffness”?
  • Please define “INR” at line 54, and CT at line 55
  • What are you measuring at line 69: (LV, 1.4 cm)? it is not clear if you performed 2 ECHOES, since in the figure legend both measured walls are 1.2 cm. In addition ECHO is described twice, at line 69 and at line 88.
  • Line 134 “may the”: a “be” is missing
  • Reviewers did not receive the supplementary material
  • Line 162: the sentence about CHF is interrupting the flow on hepatic problems.
  • The abstract sentence: “Biopsy revealed signs of drug-induced injury, consequences of subcompensated heart failure and hemosiderin accumulation” is not clear: do you mean that there are three different phenomena or that the drug injury is a consequence of subcompensated heart failure and hemosiderin accumulation? Please rephrase.
  • What do you mean at line 73: “diffuse accumulation of contrast”? since LGE is described one line after.
  • Please revise the sentence in line 171.

Round 2

Reviewer 2 Report

  • Please add the lack of endomyocardial biopsy as the limitation of the study
  • Please cite the guidelines: doi:10.1093/eurheartj/ehab368, when you describe HF, for instance at the end of page 1.
  • Scale bar of figure 3 has not been fixed, and should be fixed by the authors, not by the editor
  • when I asked to make the figures more legible even for not-experts (by adding arrows or indications of measures) you only added ovals indicating hemosiderin accumulation in liver tissue. However, this Journal is for specialists in genetics, all the rest need to be specified since you may be read by non-cardiologists, and non-epathologists.

For each figure, you need to show what you mention in the legend: for example, for Figure 2 where you notice the AV block, an example of premature supraventricular beat, RBBB combined with left anterior fascicular block, Signs of enlargement of both atria. The same applies for the other figures.

For the same reason it is important to define symbols that may be normal for pathologists, but are not of the general readier: No definition of (2+), (+1) has been given, nor in the text nor in the figure 3 legend. It is sufficient to report what was written in the reply to the reviewer: With the number of pluses our morphologists reflect the intensity of a sign.

  • Please rephrase this sentence at page 5 in the results:

In the HFE1 gene, the typical for hemochromatosis mutation in heterozygous state was found:

to

In the HFE1 gene, a recessive mutation, typically associated to hemochromatosis was found, but in heterozygous state:
